# Synthesis and Mechanical Characterization of a Ti(C,N)/Mo–Co–Ni/CaF_2_@Al_2_O_3_ Self-Lubricating Cermet

**DOI:** 10.3390/ma12233981

**Published:** 2019-11-30

**Authors:** Chuanhao Li, Mingdong Yi, Gaofeng Wei, Chonghai Xu

**Affiliations:** School of Mechanical and Automotive Engineering, Qilu University of Technology (Shandong Academy of Sciences), Jinan 250353, China; lichuanhaopaper@126.com (C.L.); weigaofeng@126.com (G.W.); xch@qlu.edu.cn (C.X.)

**Keywords:** self-lubricating cermet, multilayer core–shell microstructure, mechanical properties, residual stress

## Abstract

In this paper, an Al_2_O_3_ coated CaF_2_ (CaF_2_@Al_2_O_3_) nanocomposite powder is used as the additive phase of a Ti(C,N)-based self-lubricating cermet material. A novel self-lubricating ceramic material with a multilayer core–shell microstructure was prepared using a vacuum hot-pressing sintering process. The results show that the surface of the CaF_2_ powder is coated with Al_2_O_3_, and when introduced into a Ti(C,N)–Mo–Co–Ni material system, it can utilize the high-temperature liquid phase diffusion mechanism of the metal Mo–Co–Ni phase in the sintering process. The CaF_2_@Al_2_O_3_@Mo–Co–Ni multilayer core–shell microstructure is formed in the material. Compared with the direct addition of CaF_2_ and Al_2_O_3_, the hardness and fracture toughness of the material are increased by 24.31% and 22.56%, reaching 23.93 GPa and 9.94 MPa·m^1/2^, respectively. The formation of the multilayer core–shell microstructure is the main reason for the improvement of the mechanical properties of the material.

## 1. Introduction

Ti(C,N)-based cermet is a new type of cermet with a high hardness, high strength, good high temperature wear resistance, good toughness, low density, and high thermal conductivity [1]. The successful replacement of tungsten-based hard alloys in many applications is of great significance for strategically saving the resources of rare metal tungsten [2].

Relevant scholars have done a lot of research on improving the performance of cermet [3,4,5]. Xiang et al. [6] prepared a Ti(C_0.5_N_0.5_) powder with a particle size of less than 100 nm using the sol–gel method. Monteverde et al. [7] studied the synthesis of fine pure Ti(C,N) powders using nano-TiN and C powders as raw materials. Qu et al. [8] showed that liquid Ni–Co has a good wettability between the hard phase and the bonded phase during high temperature sintering. The results of Deng et al. [9] show that β-Co particles can optimize the microstructure of cermet and obtain excellent mechanical properties. Wu et al. [10] studied the effects of different proportions of rare metal cobalt contents on the microstructure of Ti(C,N)-based cermet materials. The results show that the core thickness of the hard particles change when Ni is replaced by Co. Xu et al. [11] showed that increasing the Ni and Co binder phase content can not only improve the solid solution reaction and the wetting of the hard phase, but also increase the thickness of the ring phase. Zhang et al. [12] studied the effect of sintering temperature on the microstructure of Al_2_O_3_-Ti(C,N)-cBN cermet. The results show that the suitable sintering temperature of Al_2_O_3_–Ti(C,N)–cBN is 1400~1500 °C [13].

Nano-particles toughen, that is, nano-particles are added into cermet to achieve the effect of strengthening and toughening by using fine-grain strengthening, bridging, a pulling-out effect, and crack deflection [14]. In order to improve the performance of ceramic materials, nano-CaF_2_ has been studied by a large number of scholars [15]. However, the application of nano-CaF_2_ is limited because of its high coefficient of thermal expansion [16]. As a result of the high thermal expansion coefficient of calcium fluoride, larger residual stress is generated [17].

In recent years, core–shell composites prepared using coating powders have been widely used in the preparation of various composite materials, such as ceramics or cemented carbides [18]. Han et al. [19] prepared porous alumina ceramics using C@Al_2_O_3_ microspheres as a pore former. Zhao et al. [20] successfully prepared core–shell composite microspheres of a nickel core and alumina nanosheet shell using the hydrothermal deposition method, and the electromagnetic absorption performance of the core–shell microspheres was significantly improved. Han et al. [21] prepared a nano-TiO_2_@Fe micro-scale photocatalyst with a core–shell structure using the precipitation method, which significantly improved the photocatalytic activity of the material. Wang et al. [22] used aluminum hydroxide to coat Eg particles in order to form a composite particle (Eg@ATH) with a core–shell structure, in order to improve the flame-retardant properties of expanded graphite in rigid polyurethane foam.

The aim of this study was to create a Ti(C,N)-based cermet material with a multilayer core–shell microstructure (CaF_2_@Al_2_O_3_@Mo–Co–Ni), and to evaluate it regarding its influence on the mechanical properties.

## 2. Experiment

### 2.1. Sample Preparation

Distilling water, xylene, and absolute ethanol were mixed at a volume ratio of 1:2:7 to obtain a mixed solution. The weighed Ca(NO)_3)_ and NH_4_F were separately dispersed in the above mixed solution to obtain a Ca(NO)_3_ solution and NH_4_F solution. Polyvinylpyrrolidone (PVP) was mixed with absolute ethanol to obtain a suspension. The suspension was added to the above Ca(NO)_3_ solution and NH_4_F solution in equal amounts, respectively, and was then ultrasonically dispersed and mechanically stirred to obtain a final Ca(NO)_3_ solution and NH_4_F solution. The Ca(NO)_3_ solution was slowly added to the NH_4_F solution, sonicated, and stirred at 70~80 °C for 20~30 min, then cooled, centrifuged, and washed to obtain nano-CaF_2_.

A uniform thickness Al(OH)_3_ coating layer was formed on the surface of the CaF_2_ powder by controlling the Al^3+^ concentration of the solution, the reaction time, the water bath temperature, and the pH of the solution after the reaction. The Al(NO_3_)_3_·9H_2_O, Polyethylene glycol (PEG, MW 6000), and nano-CaF_2_ powders were mixed. The mixed solution was subjected to mechanical stirring and ultrasonic dispersion so as to obtain a suspension of the desired solubility of the experiment. When the solution was heated to 70 °C, the NH_3_·H_2_O was slowly dropped until the pH was adjusted to the setting value (7.5). Stirring was continued for 1 h in order to complete the reaction. At this time, the Al(OH)_3_ colloid gradually coated the nano-CaF_2_. The solution was centrifuged and washed to obtain the CaF_2_@Al(OH)_3_ coated powder. The CaF_2_@Al(OH)_3_ coated powder was put into a vacuum drying oven (DZF-6050) for 12 h. The dried CaF_2_@Al(OH)_3_ coated powder was placed in a YFX5/13Q-YC box type electric resistance furnace and calcined at 1000 °C. Finally, a core–shell coated powder (CaF_2_@Al_2_O_3_) with an outer layer of Al_2_O_3_ and a core of CaF_2_ was prepared.

Ti(C,N) was used as the matrix material, and the selected Ti(C,N) had an average particle diameter of 0.5 μm and a purity of more than 99.9%. Molybdenum, cobalt, and nickel were used as the metal bonding phases, and the selected average particle diameters were 1~3 μm and the purities were 99.9%. The MgO sintering aid had a selected average particle diameter of 1 μm and a purity of 99.9%. The nano-Al_2_O_3_ had a selected average particle diameter of 500 nm and a purity of 99.9%. The CaF_2_@ Al_2_O_3_ coated powder was obtained by the experiments, and had an average particle diameter of 50 nm. Three kinds of cermet materials were prepared, according to the condition that only the nano-CaF_2_ and the nano-Al_2_O_3_ were added without adding the coated powder, and the powder was coated with CaF_2_@Al_2_O_3_ in a volume ratio of 10%. The mixture was subjected to wet ball milling, uniformly dispersed by ultrasonic dispersion, vacuum dried, and sieved through 120 mesh. The mixed powder was placed in a tubular graphite mold and underwent cold pressing (20MPa) for 30 min in order to compact the powder to reduce the pores, and then the mold was placed in a sintering furnace for vacuum hot-pressing sintering. The sintering temperature was 1450 °C, the sintering pressure was 30 MPa, and the holding time was 30 min. Finally, the Ti(C,N)/Mo–Co–Ni/10 vol %CaF_2_@Al_2_O_3_ cermet material with a multilayer core–shell microstructure was prepared.

### 2.2. Testing and Characterization

The cross-sectional microstructure of the obtained cermet material was observed using a scanning electron microscope (SEM; Hitachi Regulus8220, 5 kV, 30,000×, Tokyo, Japan) and high-resolution transmission electron microscope (HRTEM; FEI TecnaiG2F20, 200 kV, 50,000×, Hillsboro, America). The phase composition of the coated powder and the cermet material was analyzed by X-ray diffraction (XRD; XRD-6100, Co, 3 kw, 2θ: 10~90°, Kyoto, Japan) and an energy dispersive spectrometer (EDS; Hitachi S4800, 20 keV, 500×, Tokyo, Japan). The morphology and phase compositions of the CaF2@Al2O3 coated powder were analyzed using a transmission electron microscope (TEM; JEOL JEM2100, 200 kV, 40,000×, Tokyo, Japan) and scanning electron microscope (SEM; Hitachi Regulus8220, 10 kV, 6000×, Tokyo, Japan).

The prepared Ti(C,N)-based cermet disk was cut with an inner circular slicer (J5060C-1, Beijing, China), and then ground and polished with a diamond abrasive paste to finally prepare a standard test strip of 3 mm × 4 mm × 30 mm. The bending strength of the spline was measured by a three-point bending method using an electronic universal testing machine (AGS-X5KN, Kyoto, Japan), with a span of 20 mm and a loading rate of 0.5 mm/min. The hardness was measured using an Hv-120 Vickers hardness tester, and the pressurization load was 196 N and the pressure was maintained for 15 s. The indentation morphology was observed using a super depth of field three-dimensional observation microscopy system (VHX-5000, Osaka, Japan), and the two indentations and the length of the indentation diagonal were measured and recorded. The relative density of the cermet material was tested and calculated using the Archimedes drainage method.

## 3. Results and Discussion

### 3.1. Phase Analysis of the Coated Powder

Figure 1 shows the transmission electron micrograph of the nano-CaF_2_ particles. The experimental results show that the nano-CaF_2_ particles were spherical particles with a particle size of 10~20 nm, and the surface morphology of the particles was clear and complete. During the experiment, PVP was not sensitive to the pH value, and as a dispersing agent, it had less influence on the experimental process and a better dispersion effect [23]. In addition, the preparation of nano-CaF_2_ was carried out in pure water, and the calcium fluoride had different particle size [24]. If the experiment had been carried out in a hydroalcoholic mixed solution, the prepared nano-CaF_2_ would be small and severely agglomerated [25]. Therefore, a mixed solution of xylene, water, and ethanol was used as a solvent for preparing the CaF_2_. The experimental results show that the prepared nano-CaF_2_ had a moderate particle size, and the dispersion effect of the nano-CaF_2_ particles was better. Nano-CaF_2_ particles with a moderate particle size were obtained, and the excessive agglomeration of the nanoparticles was effectively avoided.

Figure 2 shows the transmission electron micrograph of the prepared CaF_2_@Al(OH)_3_ coated particles. A non-uniform nucleation method was used to control the concentration of the coating, so that the Al^3+^ concentration was between homogeneous nucleation and heterogeneous nucleation. By controlling the liquid environment and the process of nucleation and growth, it was possible to obtain a coated powder in which the coating layer was precisely controlled. As shown in the figure, the Al(OH)_3_ colloidal molecules were uniformly and stably adsorbed on the surface of the CaF_2_ nanoparticles to form CaF_2_@Al(OH)_3_ coated particles with a core–shell microstructure.

Figure 3a shows the scanning electron micrograph of the nano-CaF_2_ particle powder. We could clearly observe the surface morphology of the untreated nano-CaF_2_ granules. The surface of the nano-CaF_2_ particles was completely smooth and uniform in particle size, and was approximately spherical. Through the EDS map analysis, there were only Ca and F elements on the surface of the particles, and there was no impurity element, indicating that the prepared CaF_2_ particles had a high purity. Figure 4a shows that a dense coating layer was formed on the surface of the CaF_2_. The surface of the coating layer was composed of a large number of spherical particles, and the spherical particles were stacked and connected to each other. Through the Figure 4b analysis of the elements, compared with Figure 3b, the coated particles surface contained not only Ca and F elements, but also higher Al and O elements. In addition, the content of the Ca and F elements was significantly lower than that of the CaF_2_ particles, indicating that a large amount of Al_2_O_3_ was coated on the surface of CaF_2_, and the Al_2_O_3_ coating layer was uniform and complete.

Figure 5a shows the X-ray diffraction pattern of the nano-CaF_2_ prepared in the above experiment. As can be seen from Figure 5a, the diffraction peak was completely a characteristic peak of CaF_2_, and no impurity peak appeared, indicating that the CaF_2_ nanopowder had a good purity. The characteristic of the powder to conform to the crystal structure of fluorite, and the use of the dispersant did not affect the crystal structure of CaF_2_. In addition, the CaF_2_ spectrum not only had a sharp peak shape, but also a high intensity, indicating that its crystallinity was good. Al(OH)_3_ was formed on the surface of CaF_2_ by the heterogeneous nucleation method, and the fully reacted CaF_2_@Al(OH)_3_ suspension was placed in a vacuum drying oven and dried at 80 °C. After drying for 24 h, the CaF_2_@Al(OH)_3_ coated powder was obtained. Figure 5b shows an X-ray diffraction pattern of the CaF_2_@Al(OH)_3_ coated powder. It can be seen from the figure that the intensity of the diffraction peak was significantly reduced after the CaF_2_ was coated. As Al(OH)_3_ is a colloid and does not have a crystal structure, there was no phase of Al(OH)_3_. The CaF_2_@Al(OH)_3_ powder was calcined at a high temperature of 800 °C. Figure 5c shows the powder X-ray diffraction spectrum obtained after the calcination was completed. It can be clearly observed that the diffraction peak of CaF_2_ was sharper and the peak intensity was further enhanced. This indicates that high-temperature calcination improves the crystallinity of CaF_2_. In comparison with the CaF_2_ diffraction pattern in Figure 5a, it can be observed that the diffraction peak of Al_2_O_3_ started to appear in the spectrum of the coated powder, and there was no peak for the other phases. It was in good agreement with the CaF_2_ standard map, indicating that high-temperature calcination caused Al(OH)_3_ to form Al_2_O_3_. Figure 5d shows that the CaF_2_@Al(OH)_3_ powder was calcined at a high temperature of 1000 °C, and was subjected to X-ray diffraction analysis. Compared with the diffraction pattern of Figure 5c, it is clearly observed that the Al_2_O_3_ diffraction peak was significantly enhanced. The diffraction peak was completely a characteristic peak of α-Al_2_O_3_, and the CaF_2_@Al(OH)_3_ coated powder was calcined and finally converted into the CaF_2_@Al_2_O_3_ coated powder. In addition, the peak of CaF_2_ was sharper, and the crystallinity of CaF_2_ was enhanced, which helps to improve the self-lubricating effect of the cermet material. In addition, the CaF_2_@Al_2_O_3_ coated powder did not produce other substances during the high-temperature calcination process, indicating that the chemical compatibility of CaF_2_ and Al_2_O_3_ was better.

### 3.2. Metal Ceramic Material Microstructure

Figure 6a shows the fracture morphology of the Ti(C,N)/Mo–Co–Ni cermet material. We could clearly observe the grain shape of the material matrix, but the surface was covered with a metal binder phase. The results show that the fracture occurred at the grain boundary, and mainly expanded inside the metal bonded phase. Figure 6b shows the fracture morphology of the Ti(C,N)/Mo–Co–Ni/10 vol %CaF_2_@Al_2_O_3_ cermet material. Because of the addition of a certain amount of CaF_2_@Al_2_O_3_ coated powder, the microstructure of the multilayer core–shell can be clearly observed from the position marked in the figure. Not only was the metal bond coated with Al_2_O_3_, but nano-CaF_2_ particles were also found inside the Al_2_O_3_. Finally, the nano-CaF_2_ particles were used as the core, and the Al_2_O_3_ coated on the outer layer of CaF_2_ was used as the intermediate layer to form the multilayer core–shell microstructure of the Mo–Co–Ni metal bond phase in the outer layer of Al_2_O_3_. In addition, the step formed after the grain breakage can be clearly observed in Figure 6b, and after the grain breakage, we could observe the clear and complete nano-CaF_2_ particles. It has been indicated that there were both intergranular and transgranular fractures after the cermet material fracture. Figure 6c shows the fracture morphology of the Ti(C,N)/Mo–Co–Ni/CaF_2_/Al_2_O_3_ cermet material. As the CaF_2_ particles were not coated with Al_2_O_3_, CaF_2_ was not found in the crystal. This type of cermet was not found to form a multilayer core–shell microstructure.

Figure 7a shows an HRTEM photograph of the Ti(C,N)/Mo–Co–Ni/10 vol%CaF_2_@Al_2_O_3_ self-lubricating cermet material. We can clearly see the spherical nanoparticles in the figure. Where (b) is an enlarged picture of the selected area in (a), and (c) is an enlarged picture of the selected area in (b). The results of the high-resolution transmission electron microscopy show that the nanoparticles in the crystal were CaF_2_ crystals with a typical crystal structure of fluorite, and the particle size was about 10 nm, which is basically consistent with the particle size of the coated nano-powder prepared above. In addition, it could be clearly observed that the nanoparticles did not agglomerate, and the nano-CaF_2_ particles did not grow abnormally during the hot pressing sintering. The CaF_2_ particles still ensured a good spherical particle shape in the matrix material. The nano-CaF_2_ particles of the core inside and the outer shell of the Al_2_O_3_ coating formed a stable core–shell structure. This core–shell structure provided a prerequisite for obtaining the excellent mechanical properties of Ti(C,N)-based cermet materials. The CaF_2_ crystal index was (200), and according to the HRTEM photo analysis, the interplanar spacing of the intra-cavity CaF_2_ crystal was 0.189 nm, which is slightly lower than the standard interplanar spacing of 0.193 nm for the CaF_2_ crystal. The reason for the analysis was that the growth of the nano-Al_2_O_3_ coated on the surface of CaF_2_ began at 1200 °C [26] during the sintering process. The growth of the CaF_2_ crystal was confined within the Al_2_O_3_ crystal, and its coefficient of thermal expansion was high, but the modulus of elasticity was small. Therefore, the lattice spacing was reduced from the theoretical value.

Figure 8 shows the scanning electron micrograph of the indentation crack of the Ti(C,N)/Mo–Co–Ni/10 vol%CaF_2_@Al_2_O_3_ self-lubricating cermet material. Figure 8a shows this in the form of crack deflection and crack branching. Figure 8b clearly shows the crack bridging and crack deflection morphology. The deflection of the crack was mainly due to the fact that when the crack propagated to the multilayer core–shell microstructure, the thermal expansion coefficient of the intragranular CaF_2_ was large, resulting in residual tensile stress. When the crack propagated to the CaF_2_@Al_2_O_3_ coated powder, it tended to expand toward the grain boundary. This phenomenon caused the crack to be deflected, so that the expansion path was prolonged and the fracture energy was partially consumed, thereby producing a toughening effect.

### 3.3. Mechanical Properties of Cermet Materials

The Ti(C,N)/Mo–Co–Ni and Ti(C,N)/Mo–Co–Ni/CaF_2_/Al_2_O_3_ cermet materials and coated powder addition volumes were tested according to the test methods described in the previous section. The Ti(C,N)/Mo–Co–Ni/10 vol%CaF_2_@Al_2_O_3_ cermet material had a content of 10%. The test results of the mechanical properties of the Ti(C,N)-based self-lubricating cermet materials are shown in Table 1. The experimental results show that the comprehensive mechanical properties of the Ti(C,N)/Mo–Co–Ni/10 vol%CaF_2_@Al_2_O_3_ cermet materials were the best. The hardness was 23.93 GPa, the fracture toughness was 9.94 MPa·m^1/2^, and the flexural strength was 997 MPa. Compared with the non-added CaF_2_@Al_2_O_3_ coated powder, the hardness and fracture toughness of the Ti(C,N)-based self-lubricating cermet materials increased by 24.25% and 31.83%, respectively. Compared with the direct addition of nano-CaF_2_ and nano-Al_2_O_3_, the hardness and fracture toughness of the Ti(C,N)-based self-lubricating cermet materials increased by 24.31% and 22.56%, respectively. The formation of a multilayer core–shell microstructure was the main reason for the improvement of the mechanical properties.

In future studies, the effect of the content of the CaF_2_@Al_2_O_3_ coated powder on the mechanical properties of Ti(C,N)-based cermet materials needs further experimental verification, and the formation mechanism of the multilayer core–shell microstructure needs more in-depth research.

## 4. Conclusions

(1) An advanced cermet material with a multilayer core–shell microstructure was successfully prepared using a vacuum hot-pressing sintering process. Compared with the general Ti(C,N)–Mo–Co–Ni cermet materials, the fracture toughness and hardness of the Ti(C,N)/Mo–Co–Ni/10 vol%CaF_2_@Al_2_O_3_ cermet materials are improved by 31.83% and 24.25%, respectively. A multilayer core–shell microstructure with nano-CaF_2_ as the core, Al_2_O_3_ as the intermediate layer, and a metal phase as the shell is formed.

(2) The formation of a multi-layer core–shell microstructure can effectively improve the mechanical properties of Ti(C,N)–Mo–Co–Ni cermet. The fracture mode of the material is the wear-through/inter-grain hybrid fracture, and the toughening mode is the crack deflection, crack bridging, and crack branching.

## Figures and Tables

**Figure 1 materials-12-03981-f001:**
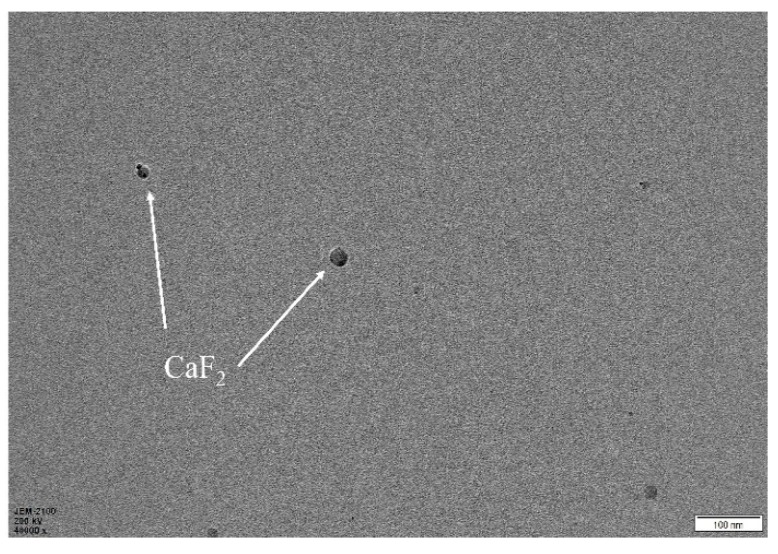
TEM photographs of nano-CaF_2_.

**Figure 2 materials-12-03981-f002:**
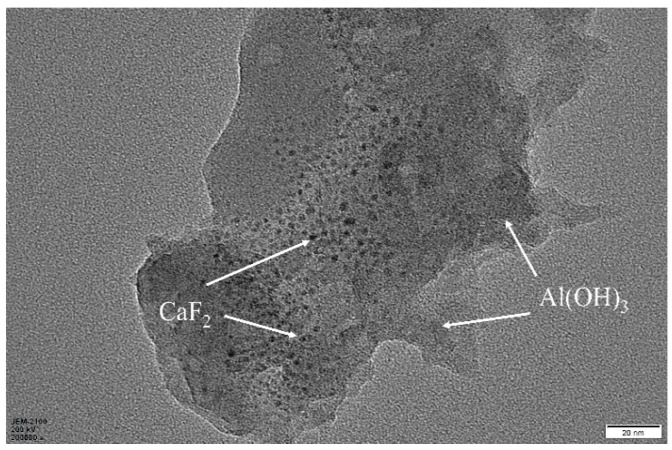
TEM photographs of CaF_2_@Al(OH)_3_.

**Figure 3 materials-12-03981-f003:**
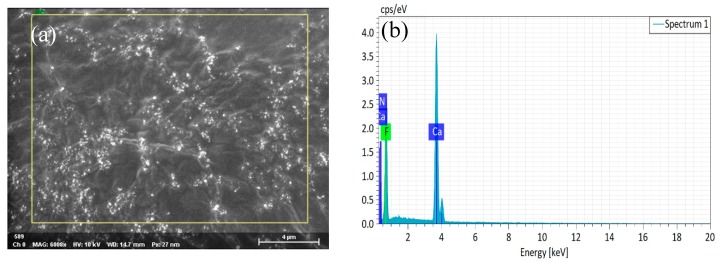
(**a**) SEM photographs of nano-CaF_2_, (**b**) energy dispersive spectrometer (EDS) spectra of nano-CaF_2_.

**Figure 4 materials-12-03981-f004:**
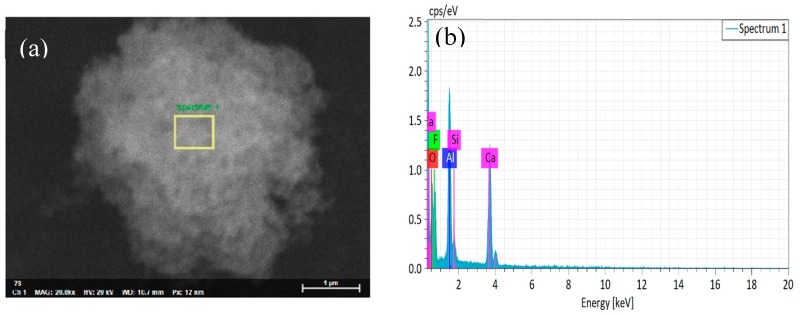
(**a**) SEM photographs of the CaF_2_@Al_2_O_3_ coated powder and (**b**) EDS spectra of CaF_2_@Al_2_O_3_ coated powder sintered at 1000 °C.

**Figure 5 materials-12-03981-f005:**
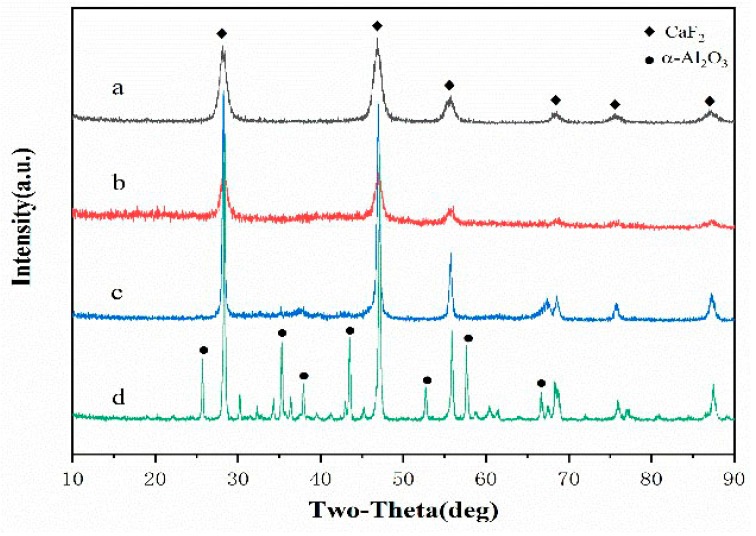
XRD patterns of powders: (**a**) CaF_2_, (**b**) CaF_2_@Al_2_O_3_ calcined at 80 °C, (**c**) CaF_2_@Al_2_O_3_ calcined at 800 °C, and (**d**) CaF_2_@Al_2_O_3_ calcined at 1000 °C.

**Figure 6 materials-12-03981-f006:**
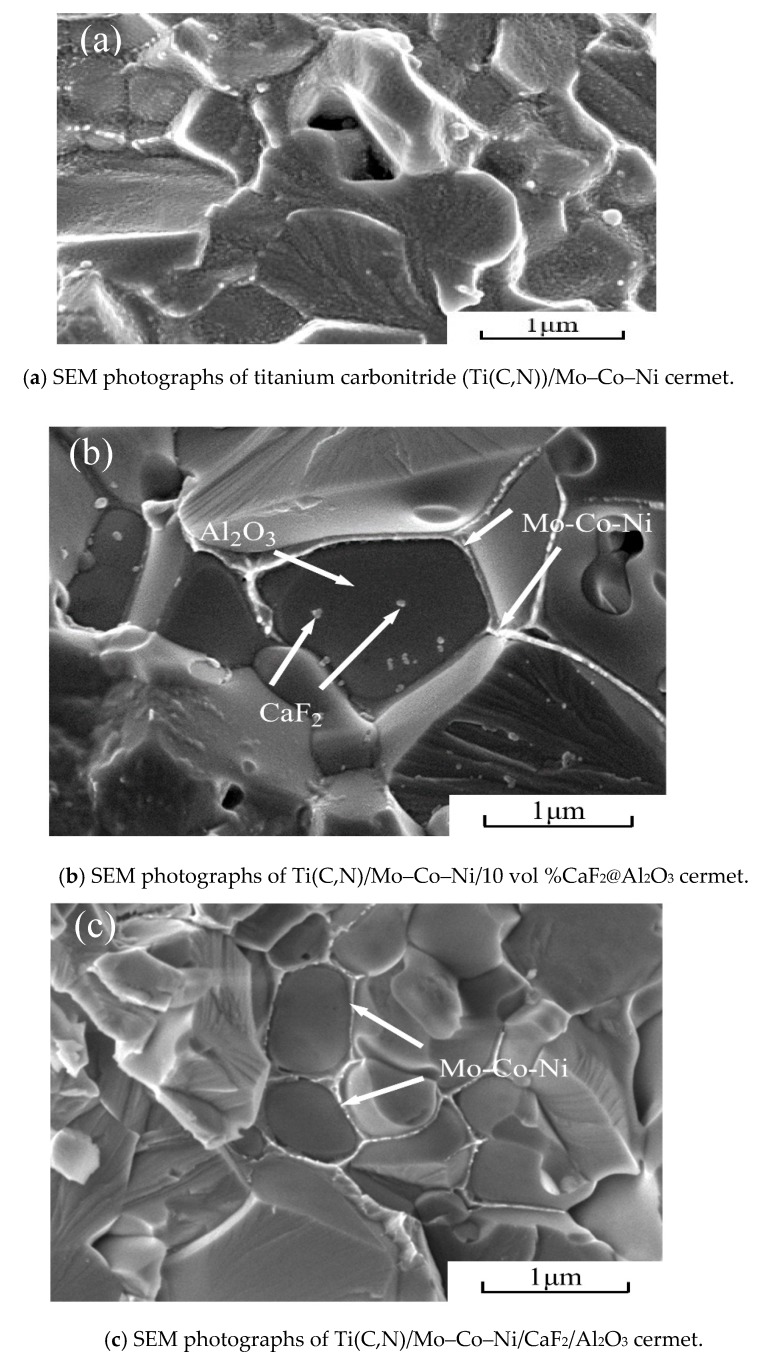
SEM photographs of cermet, (**a**) SEM photographs of titanium carbonitride (Ti(C,N))/Mo–Co–Ni cermet; (**b**) SEM photographs of Ti(C,N)/Mo–Co–Ni/10 vol %CaF_2_@Al_2_O_3_ cermet.; (**c**) SEM photographs of Ti(C,N)/Mo–Co–Ni/CaF_2_/Al_2_O_3_ cermet.

**Figure 7 materials-12-03981-f007:**
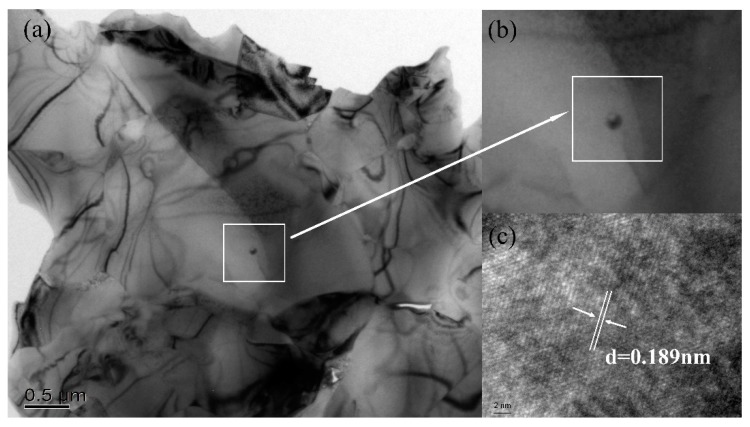
HRTEM photographs of Ti(C,N)/Mo–Co–Ni/10 vol%CaF_2_@Al_2_O_3_ cermet (**a**) HRTEM image of Ti(C,N)/Mo–Co–Ni/10 vol%CaF_2_@Al_2_O_3_ cermet; (**b**) HRTEM image of CaF_2_; (**c**) Diffraction pattern of CaF_2_.

**Figure 8 materials-12-03981-f008:**
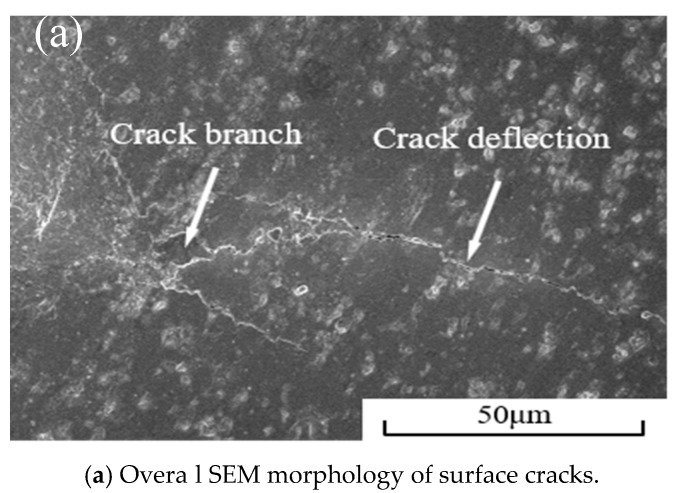
SEM photographs of Ti(C,N)/Mo–Co-Ni/10vol%CaF_2_@Al_2_O_3_ cermet surface cracks, (**a**) Overall SEM morphology of surface cracks; (**b**) SEM morphology of surface cracks enlarge.

**Table 1 materials-12-03981-t001:** Mechanical properties of cermet materials.

Component	Flexure Strength /MPa	Fracture Toughness /MPa·m^1/2^	Hardness /GPa	Relative Density /%
Ti(C,N)/Mo–Co–Ni	1055 ± 27	7.54 ± 0.14	19.26 ± 0.22	98.87 ± 0.032
Ti(C,N)/Mo–Co–Ni/CaF_2_/Al_2_O_3_	1016 ± 20	8.11 ± 0.12	19.25 ± 0.13	98.89 ± 0.033
Ti(C,N)/Mo–Co–Ni/10 vol%CaF_2_@Al_2_O_3_	997 ± 21	9.94 ± 0.23	23.93 ± 0.21	99.43 ± 0.021

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
