# Peer review of "Synthesis and Mechanical Characterization of a Ti(C,N)/Mo–Co–Ni/CaF2@Al2O3 Self-Lubricating Cermet"

_materials, 2019, doi:10.3390/ma12233981_

Round 1

Reviewer 1 Report

The reviewer has the following questions for the authors to respond:

Line 114: can the author provide statistical analysis from TEM about the size distribution. It is impossible to get 10~20 nm particle size from figure 1, as the scale is 200 nm. Will higher magnification of the TEM can be used? Line 133: one cannot get the conclusion just by looking at figure 2 that CaF2@Al(OH)3 is a core-shell microstructure. Again, this work needs a higher magnification. The plotting of TEM and SEM is not consistent. Please use the original figure from the instrument with parameters and scales on them. In the “2. Experiment” section, the authors need to provide the detailed instrument parameters used for SEM and TEM characterization. Also, no XRD instrumental information nor the data collection details. Line 228: CaF2 does not have a perovskite structure.

Editorial comments:

Subscript needs to be used for all chemical formulae.  

The authors should also check English before resubmission.

Author Response

Response to Reviewers

Dear editors and reviewers:

    Thank you for your valuable comments and suggestions on our manuscript entitled “The effect of adding CaF2@Al2O3 on the mechanical properties of Ti(C,N)-based self-lubricating cermet” (Manuscript number: Materials-629439). Your comments and suggestions are very helpful for improving our manuscript quality. We have carefully studied the comments and suggestions and made some revisions marked by the yellow backgrounds in the manuscript, and English of the whole paper has been carefully edited. I sincerely hope these revisions and responses can meet your requirements. The detailed responses to the reviewers’ comments are listed as following:

Reviewer 1#:Manuscript Number: Materials-629439

Line 114: can the author provide statistical analysis from TEM about the size distribution. It is impossible to get 10~20 nm particle size from figure 1, as the scale is 200 nm. Will higher magnification of the TEM can be used?

Response: Thanks for your comment and suggestion. According to your suggestion, we have added higher magnification of the TEM, as the scale is 100 nm. We counted the particle size of nano-particles in transmission electron microscope. The results show that the particle sizes are between 10~20nm and the average particle size is 17 nm. Line 154

Line 133: one cannot get the conclusion just by looking at figure 2 that CaF2@Al(OH)3 is a core-shell microstructure. Again, this work needs a higher magnification. The plotting of TEM and SEM is not consistent. Please use the original figure from the instrument with parameters and scales on them. In the “2. Experiment” section, the authors need to provide the detailed instrument parameters used for SEM and TEM characterization. Also, no XRD instrumental information nor the data collection details.

Response: Thanks for your comment and suggestion. According to your suggestion, we have added higher magnification of the TEM, as the scale is 20 nm. We have adjusted the TEM and SEM images. We use heterogeneous nucleation principle to coat Al(OH)3 colloid on the surface of CaF2. After high temperature sintering, Al(OH)3 colloid is completely converted into α-Al2O3. As shown in Fig. 3(b) and Fig. 4(b). After CaF2 is coated. the content of Ca and F elements is significantly lower than that of CaF2 particles, indicating that a large amount of Al2O3 is coated on the surface of CaF2. In the “2. Experiment” section, we have added the instrument parameters. The revisions was marked by the yellow backgrounds in the manuscript. Line 104

Line 228: CaF2 does not have a perovskite structure.

Response: Thanks for your comment and suggestion. According to your suggestion, we have changed “perovskite structure” to “crystal structure of fluorite”. Line 207

Reviewer 2 Report

In this study, Al2O3 coated CaF2(CaF2@Al2O3) nanocomposite powder is used as the additive phase of Ti(C,N)-based self-lubricating cermet material to improve its mechanical properties. Before the manuscript can be accept for publication, major revisions are necessary considering the following comments.

Format and spelling mistakes appear in the paper, the authors should check carefully and correct them. Introduction part should be more focus on related problem that the authors wish to solve. Line 77, ‘PH’ should be ‘pH’. Line 90, what is the pressure of cold pressing? Why does it take 30 min to press? Line 147, EDS analysis cannot indicate that Al2O3 coating is uniform and complete. Fig.6(b) and (c) cannot be found. Line 220, why is not CaF2 found in the crystal? Is CaF2 able to react with the metal binder? Fig.7, why could only one nanoparticle be found when 10vol% CaF2@Al2O3 is added? EDS test is suggested to clarify the composition of the nanoparticle found in the cermet material. In addition, HRTEM image should be identified with crystal indices and according interplanar spacings. Line 223, “Table 10. 6” should be “Figure 6”; Fig.8, title of (a) and (b) is the same. Line 249, the format of this paragraph seems to be wrong. Table 1, what is the theoretical density used when calculating relative density? Line 273, conclusion 1 should be rewritten. Line 60, the authors mention that the introduction of CaF2 can significantly improve self-lubrication properties. No evidence can be found in this paper to support this. Line 62, the authors mention that “the multilayer core-shell structure formed by the double coating of Al2O3 and the metal layer on CaF2 effectively improves the residual stress”, while there is no evidence in this paper to support this.

Author Response

Response to Reviewers

Dear editors and reviewers:

    Thank you for your valuable comments and suggestions on our manuscript entitled “The effect of adding CaF2@Al2O3 on the mechanical properties of Ti(C,N)-based self-lubricating cermet” (Manuscript number: Materials-629439). Your comments and suggestions are very helpful for improving our manuscript quality. We have carefully studied the comments and suggestions and made some revisions marked by the yellow backgrounds in the manuscript, and English of the whole paper has been carefully edited. I sincerely hope these revisions and responses can meet your requirements. The detailed responses to the reviewers’ comments are listed as following:

Reviewer 2#:Manuscript Number: Materials-629439

In this study, Al2O3 coated CaF2(CaF2@Al2O3) nanocomposite powder is used as the additive phase of Ti(C,N)-based self-lubricating cermet material to improve its mechanical properties. Before the manuscript can be accept for publication, major revisions are necessary considering the following comments.

Format and spelling mistakes appear in the paper, the authors should check carefully and correct them. Introduction part should be more focus on related problem that the authors wish to solve.

Response: Thanks for your comment and suggestion. According to your suggestion, we have corrected format and spelling mistakes appear in the manuscript. We added some content in the introduction part. Line 57

Line 77, ‘PH’ should be ‘pH’.

Response: Thanks for your comment and suggestion. According to your suggestion, we have changed ‘PH’ to ‘pH’. Line 81

Line 90, what is the pressure of cold pressing? Why does it take 30 min to press?

Response: Thanks for your comment and suggestion. According to your suggestion, we have added the pressure of cold pressing. The pressure of cold pressing is approximately 20 MPa. By cold pressing at 20 MPa for 30 min, the excess gas in the mold is removed, and the green body is molded to increase the density of the material. Line 98

Line 147, EDS analysis cannot indicate that Al2Ocoating is uniform and complete. Fig.6(b) and (c) cannot be found.

Response: Thanks for your comment and suggestion. According to your suggestion, we have changed the EDS spectra, as shown in Fig. 3(b) and Fig. 4(b). After CaF2 is coated, the content of Ca and F elements is significantly lower than that of CaF2 particles, indicating that a large amount of Al2O3 is coated on the surface of CaF2. Thank you very much for pointing out our mistake. We have added Fig. 6(b) and Fig. 6(c). Line 172

Line 220, why is not CaF2 found in the crystal? Is CaF2 able to react with the metal binder? Fig.7, why could only one nanoparticle be found when 10vol% CaF2@Al2O3 is added? EDS test is suggested to clarify the composition of the nanoparticle found in the cermet material. In addition, HRTEM image should be identified with crystal indices and according interplanar spacings.

Response: Thanks for your comment and suggestion. We have added Fig. 6(b) and Fig. 6(c). As shown in Fig 6(b), we can observe the CaF2 in the crystal. The reaction of CaF2 with metal binder requires further investigation. The cermet materials are cut, ground and polished into several 3 mm×4 mm×0.5 mm thin slice. The thin slice is ground to a thickness of 75 μm using a diamond abrasive paste. Then the thickness of the slice is reduced to 10⁓30 μm using an ion beam thinner. Since the sample is relatively thin, fewer nanoparticles are observed. We have added interplanar spacings in Fig. 7. The CaF2 crystal indices is ( 2 0 0 ). Line 327

Line 223, “Table 10. 6” should be “Figure 6”; Fig.8, title of (a) and (b) is the same.

Response: Thanks for your comment and suggestion. According to your suggestion, we have changed “Table 10. 6” to “Figure 6”. According to your suggestion, we have changed Fig. 8 caption. Line 390

Line 249, the format of this paragraph seems to be wrong. Table 1, what is the theoretical density used when calculating relative density?

Response: Thanks for your comment and suggestion. According to your suggestion, we have corrected format. The theoretical density is 5.15 g/cm3.

Line 273, conclusion 1 should be rewritten.

Response: Thanks for your comment and suggestion. According to your suggestion, we have rewritten the conclusion 1. Line 410

Line 60, the authors mention that the introduction of CaF2 can significantly improve self-lubrication properties. No evidence can be found in this paper to support this.

Response: Thanks for your comment and suggestion. According to your suggestion, we have added some references to the manuscript. Many researchers (including our research group) have carried out a large number of CaF2 dry friction performance studies. As shown in the references:

[1] Jianxin D, Tongkun C, Xuefeng Y, et al. Self-lubrication of sintered ceramic tools with CaF2 additions in dry cutting[J]. International Journal of Machine Tools and Manufacture, 2006, 46(9): 957-963.

The composition of the self-lubricating film was found to be mainly CaF2, which released and smeared on the wear track of the tool rake face between the tool-chip sliding couple during machining processes.

[2] Piasecki A, Kulka M, Kotkowiak M. Wear resistance improvement of 100CrMnSi6-4 bearing steel by laser boriding using CaF2 self-lubricating addition[J]. Tribology International, 2016, 97: 173-191.

Tribofilm, consisting of CaF2, was observed on the worn surface and was the reason for improved wear resistance compared to the laser-alloyed layer with boron only.

[3] Kong L, Zhu S, Bi Q, et al. Friction and wear behavior of self-lubricating ZrO2 (Y2O3)-CaF2-Mo-graphite composite from 20 ℃ to 1000 ℃[J]. Ceramics International, 2014, 40(7): 10787-10792.

The low friction coefficients of ZrO2 (Y2O3)-Mo-CaF2-graphite at high temperatures were due to the perfect lubricity of CaMoO4 which was formed on the worn surface at high temperatures.

[4] Wu G, Xu C, Xiao G, et al. Self-lubricating ceramic cutting tool material with the addition of nickel coated CaF2 solid lubricant powders[J]. International Journal of Refractory Metals and Hard Materials, 2016, 56: 51-58.

The as-prepared self-lubricating ceramic cutting tool material made by adding nickel coated CaF2 powders exhibited notable improvements in microstructure and mechanical properties, in comparison with the corresponding cutting tool material made by directly adding uncoated CaF2 powders.

The addition of CaF2 powder to ceramic materials can significantly reduce the coefficient of friction of the material. Nevertheless, the addition of low mechanical properties of CaF2 powder reduces the mechanical properties of the ceramic material. Therefore, we introduce nano-CaF2 into the interior of the Al2O3 crystal. Furthermore, the multilayer core-shell microstructure is formed in the cermet material. Finally, we obtained self-lubricating cermet materials with good mechanical properties. Moreover, the HRTEM shows that the crystal structure of nano-CaF2 in the Al2O3 crystal has not been destroyed and its self-lubricating property is retained. The friction research of TMC( Ti(C,N)/Mo-Co-Ni/CaF2@Al2O3 ) is in progress. We used TMC and Si3N4 ceramic balls for dry friction test. The results show that adding 10% CaF2@Al2O3 powder to cermet material reduces its friction coefficient from 0.5⁓0.8 to 0.3⁓0.4 under the friction conditions of 20⁓140 m/min and 5⁓20 N.

Line 62, the authors mention that “the multilayer core-shell structure formed by the double coating of Al2O3 and the metal layer on CaF2 effectively improves the residual stress”, while there is no evidence in this paper to support this.

Response: Thanks for your comment and suggestion. We have added Fig. 6(b) and Fig. 6(c). As shown in Fig 6(b), we can observe the CaF2 in the crystal. The multilayer core-shell microstructure is shown in Fig 6(b). Heterogeneous nucleation principle is employed for the preparation of CaF2@Al2O3. Through ultrasonic dispersion and mechanical ball grinding, CaF2@Al2O3 coated powder and metal powder are mixed evenly. With the increase of sintering temperature, nano-Al2O3 grains began to grow. The metal powder melt around the CaF2@Al2O3 coated powder. Finally, the liquid phase metal is diffused on the surface of the CaF2@Al2O3 coated powder. The liquid phase metal inhibited the growth of nano-Al2O3 grain and refined the grain. As shown in Fig 6(b), the multilayer core-shell microstructure with nano-CaF2 as the core, Al2O3 as the intermediate layer and metal phase as the shell is formed. Line 286

Reviewer 3 Report

Please add the number of the affiliation to each author and provide contact details for the corresponding author (see instructions for authors)

The introduction lacks references in several statements. Please add references at lines 22, 24, 42, 44

Please put et al. in italic form and add a “.” after al.

The introduction needs to be rewritten. Instead of describing different studies one by one, the authors need to provide a review of the important information in the topic of the manuscript, supported by the literature.

What lines 55-56 mean?

Lines 56-64 describe methodology and results. It must be removed from the introduction section

What is the aim of the work? A paragraph referring to this needs to be added to the manuscript

Lines 75-79 provide a summary of the methodology, described in detail in lines 80-94. Do authors think this information needs to be repeated?

Line 95: please put the subtitle in italic form, as the others

Lines 96-100: please give in full the meaning of SEM, HRTEM; EDS; TEM first time you use the terms. Abbreviations must only be used, after the terms appear in full the first time

Did the authors perform a statistical analysis of the obtained results? This is fundamental because without this it can not be concluded an improvement of the material properties

Please replace all sentences referring to “figure X is” by “figure X shows”

Line 116: PVP already appeared referred at line 69, so just use the abbreviation

Figure 3: Please refer to figure 3a and figure 3b in the text. Add that information in the figure. 

Figure 4: Please refer to figure 4a and figure 4b in the text. Add that information in the figure. 

Figures need to be repositioned in the manuscript, to be near the respective text

Line 147: the authors refer that "In addition, the content of Ca and F elements is significantly lower than that of CaF2 particles, indicating that a large amount of Al2O3 is coated on the surface of CaF2, and the Al2O3 coating layer is uniform and complete.” Please explain how this can be concluded from the EDS spectra.

Figure 5 caption needs to be changed, since (a) refers to CaF2 power and not CaF2@Al2O3. 

Figure 6 is wrongly captioned as table 10, please change

Lines 207-210: authors refer to figure 6a and 6b, however, figure 6 does not present a and b. Please add the correct figure.

Line 224 states that figure 7 shows the HRTEM photograph of a Ti(C,N)/Mo-Co-Ni/10vol%CaF2@Al2O3 material, however, figure 7 caption only refers to Ti(C,N) cement. Please change it.

Figure 8 caption presents information that belongs to results and discussion and not to the figure caption. Please change it.

All figure captions need to be improved since they do not provide sufficient information. Please see the published papers in the Materials journal for guidance.

Please add a table 1 information about statistical differences between the obtained results

References are not following journal guidelines. Please change it.

Author Response

Response to Reviewers

Dear editors and reviewers:

    Thank you for your valuable comments and suggestions on our manuscript entitled “The effect of adding CaF2@Al2O3 on the mechanical properties of Ti(C,N)-based self-lubricating cermet” (Manuscript number: Materials-629439). Your comments and suggestions are very helpful for improving our manuscript quality. We have carefully studied the comments and suggestions and made some revisions marked by the yellow backgrounds in the manuscript, and English of the whole paper has been carefully edited. I sincerely hope these revisions and responses can meet your requirements. The detailed responses to the reviewers’ comments are listed as following:

Reviewer # 3: Manuscript Number: Materials-629439

Please add the number of the affiliation to each author and provide contact details for the corresponding author (see instructions for authors)

Response: Thanks for your comment and suggestion. According to your suggestion, we have added the number of the affiliation to each author and provide the Email-address for the corresponding author. line 4

The introduction lacks references in several statements. Please add references at lines 22, 24, 42, 44

Response: Thanks for your comment and suggestion. According to your suggestion, we have added references at lines 22, 24, 42, 44. line 24

Please put et al. in italic form and add a “.” after al.

Response: Thanks for your comment and suggestion. According to your suggestion, we have changed the form and added punctuation. line 24

The introduction needs to be rewritten. Instead of describing different studies one by one, the authors need to provide a review of the important information in the topic of the manuscript, supported by the literature.

Response: Thanks for your comment and suggestion. According to your suggestion, we have rewritten the introduction. line 24

What lines 55-56 mean?

Response: Thanks for your comment and suggestion. we have added Fig 6(b). As is shown in Fig. 6(b), the nano-CaF2 particles were used as the core, and Al2O3 coated on the outer layer of CaF2 was used as the intermediate layer to form the multilayer core-shell microstructure of the Mo-Co-Ni metal bond phase in the outer layer of Al2O3.

Lines 56-64 describe methodology and results. It must be removed from the introduction section.What is the aim of the work? A paragraph referring to this needs to be added to the manuscript

Response: Thanks for your comment and suggestion. According to your suggestion, we have rewritten the introduction.

Lines 75-79 provide a summary of the methodology, described in detail in

Response: Thanks for your comment and suggestion. According to your suggestion, we have rewritten this paragraph. line 76

lines 80-94. Do authors think this information needs to be repeated?

Response: Thanks for your comment and suggestion. As shown in Fig 6(b), we can observe the CaF2 in the crystal. The multilayer core-shell microstructure is shown in Fig 6(b). Heterogeneous nucleation principle is employed for the preparation of CaF2@Al2O3. Through ultrasonic dispersion and mechanical ball grinding, CaF2@Al2O3 coated powder and metal powder are mixed evenly. With the increase of sintering temperature, nano-Al2O3 grains began to grow. The metal powder melt around the CaF2@Al2O3 coated powder. Finally, the liquid phase metal is diffused on the surface of the CaF2@Al2O3 coated powder. The liquid phase metal inhibited the growth of nano-Al2O3 grain and refined the grain. As shown in Fig 6(b), the multilayer core-shell microstructure with nano-CaF2 as the core, Al2O3 as the intermediate layer and metal phase as the shell is formed.

This is the detail preparation process of Ti(C,N)/Mo-Co-Ni/10vol% CaF2@Al2O3 cermet materials.

Line 95: please put the subtitle in italic form, as the others

Response: Thanks for your comment and suggestion. According to your suggestion, we have changed the form of the subtitle. line 103

Lines 96-100: please give in full the meaning of SEM, HRTEM; EDS; TEM first time you use the terms. Abbreviations must only be used, after the terms appear in full the first time

Response: Thanks for your comment and suggestion. According to your suggestion, we have added the the meaning of SEM, HRTEM; EDS. line 104

Did the authors perform a statistical analysis of the obtained results? This is fundamental because without this it can not be concluded an improvement of the material properties

Response: Thanks for your comment and suggestion. We tested each material five times in the process of mechanical property experiment.  The average value is calculated as the mechanical property value of the material. According to your suggestion, we have added relevant information in table 1.

Please replace all sentences referring to “figure X is” by “figure X shows”

Response: Thanks for your comment and suggestion. According to your suggestion, we have changed “figure X is” to “figure X shows”.

Line 116: PVP already appeared referred at line 69, so just use the abbreviation

Response: Thanks for your comment and suggestion. According to your suggestion, we have removed “polyvinylpyrrolidone”. line 126

Figure 3: Please refer to figure 3a and figure 3b in the text. Add that information in the figure. 

Response: Thanks for your comment and suggestion. According to your suggestion, we have added the mark to figure 3a and figure 3b and relevant information in the manuscript.

Figure 4: Please refer to figure 4a and figure 4b in the text. Add that information in the figure. Figures need to be repositioned in the manuscript, to be near the respective text

Response: Thanks for your comment and suggestion. According to your suggestion, we have added the mark to figure 4a and figure 4b and relevant information in the manuscript.

Line 147: the authors refer that "In addition, the content of Ca and F elements is significantly lower than that of CaF2 particles, indicating that a large amount of Al2O3 is coated on the surface of CaF2, and the Al2O3 coating layer is uniform and complete.” Please explain how this can be concluded from the EDS spectra.

Response: Thanks for your comment and suggestion. According to your suggestion, We have changed the EDS spectra. As shown in Fig. 3(b) and Fig. 4(b). After CaF2 is coated. the content of Ca and F elements is significantly lower than that of CaF2 particles, indicating that a large amount of Al2O3 is coated on the surface of CaF2.

Figure 5 caption needs to be changed, since (a) refers to CaF2 power and not CaF2@Al2O3

Response: Thanks for your comment and suggestion. Spectrum (a) refers to CaF2 power. Spectrum (b) refers to CaF2@Al(OH)3 coated powder at 80 °C. Spectrum (c) refers to CaF2@Al(OH)3 coated powder at 800 °C. Spectrum (d) refers to CaF2@Al2O3 powders at 1000 °C. The diffraction peak is completely characteristic peak of α-Al2O3, and the CaF2@Al(OH)3 coated powder is calcined and finally converted into CaF2@Al2O3 coated powder.

Figure 6 is wrongly captioned as table 10, please change

Response: Thanks for your comment and suggestion. According to your suggestion, we have changed Figure 6 caption.

Lines 207-210: authors refer to figure 6a and 6b, however, figure 6 does not present a and b. Please add the correct figure.

Response: Thanks for your comment and suggestion. According to your suggestion, we have added the correct figure.

Line 224 states that figure 7 shows the HRTEM photograph of a Ti(C,N)/Mo-Co-Ni/10vol%CaF2@Al2O3 material, however, figure 7 caption only refers to Ti(C,N) cement. Please change it.

Response: Thanks for your comment and suggestion. According to your suggestion, we have changed figure 7 caption. line 344

Figure 8 caption presents information that belongs to results and discussion and not to the figure caption. Please change it.

Response: Thanks for your comment and suggestion. According to your suggestion, we have changed figure 8 caption. line 390

All figure captions need to be improved since they do not provide sufficient information. Please see the published papers in the Materials journal for guidance.

Response: Thanks for your comment and suggestion. According to your suggestion, we have changed relevant figure captions.

Please add a table 1 information about statistical differences between the obtained results

Response: Thanks for your comment and suggestion. According to your suggestion, we have added relevant information in table 1. line 408

References are not following journal guidelines. Please change it.

Response: Thanks for your comment and suggestion. According to your suggestion, we have changed references. line 433

Round 2

Reviewer 1 Report

The authors addressed the reviewer's questions. 

Author Response

Thank you for your valuable comments and suggestions on our manuscript entitled “The effect of adding CaF2@Al2O3 on the mechanical properties of Ti(C,N)-based self-lubricating cermet” (Manuscript number: Materials-629439). Your comments and suggestions are very helpful for improving our manuscript quality. We have carefully studied the comments and suggestions and made some revisions marked by the yellow backgrounds in the manuscript.

Reviewer 2 Report

This paper is now worthy to be published.

Author Response

(The authors gave the same response as above.)

Reviewer 3 Report

Please add a reference at line 40

Line 48: replace “,” by “or” at the phrase (ceramics or cemented carbides)

Are phrases at lines 57-58 the summary of the information described from line 47 to line 56? If so, the phrase needs to be clearer.

Lines 59-65 describe methodology and results. It must be removed from the introduction section.

A clearer aim needs to be written. I suggest the authors use something like: "the aim of this study was to create a cermet material with a multilayer core-shell structure and to evaluate it regarding…"

Figure 3 and figure 4 captions need to be rewritten. The figures do not only show the spectra but also micrographs. That information needs to be added to the captions.

Figure 5 caption needs to be changed, since (a) refers to CaF2 power and not CaF2@Al2O3. 

The authors stated that: “The experimental results show that the comprehensive mechanical properties of Ti(C, N)/Mo-Co-Ni/10vol%CaF2@Al2O3 cermet materials are the best.” It is impossible to conclude this without a statistical analysis of the results. The authors assume that the values are significant just because they are higher than the other materials, but does this mean something? Is this significant?

Lines 416-417: the authors state that: “The formation of multi-layer core-shell microstructure can effectively improve the mechanical properties of Ti(C, N)-Mo-Co-Ni cermet”. Once again this cannot be concluded without a statistical analysis of the results. Just because the obtained values are higher (slightly) does not mean the material has improved properties.

A paragraph discussing the study limitations needs to be added to the manuscript.

I suggest the paper title be changed, to more clearly describe the experimental work. For example: "synthesis and mechanical characterization of a Ti(C,N)/MoCoNi/CaF2@Al2O3 self-lubricating cermet”.

Author Response

Response to Reviewers

Dear reviewers:

    Thank you for your valuable comments and suggestions on our manuscript entitled “The effect of adding CaF2@Al2O3 on the mechanical properties of Ti(C,N)-based self-lubricating cermet” (Manuscript number: Materials-629439). Your comments and suggestions are very helpful for improving our manuscript quality. We have carefully studied the comments and suggestions and made some revisions marked by the yellow backgrounds in the manuscript, and English of the whole paper has been carefully edited. I sincerely hope these revisions and responses can meet your requirements. The detailed responses to the reviewers’ comments are listed as following:

Reviewer 3#:Manuscript Number: Materials-629439

Please add a reference at line 40

Response: Thanks for your comment and suggestion. According to your suggestion, we have added the reference.

Line 48: replace “,” by “or” at the phrase (ceramics or cemented carbides)

Response: Thanks for your comment and suggestion. According to your suggestion, we have corrected the punctuation.

Are phrases at lines 57-58 the summary of the information described from line 47 to line 56? If so, the phrase needs to be clearer.

Response: Thanks for your comment and suggestion. According to your suggestion, we have corrected the phrase.

Lines 59-65 describe methodology and results. It must be removed from the introduction section. A clearer aim needs to be written. I suggest the authors use something like: "the aim of this study was to create a cermet material with a multilayer core-shell structure and to evaluate it regarding…"

Response: Thanks for your comment and suggestion. According to your suggestion, we deleted the methodology and results in the introduction and corrected the research objectives.

Figure 3 and figure 4 captions need to be rewritten. The figures do not only show the spectra but also micrographs. That information needs to be added to the captions.

Response: Thanks for your comment and suggestion. According to your suggestion, we have corrected the captions.

Figure 5 caption needs to be changed, since (a) refers to CaF2 power and not CaF2@Al2O3

Response: Thanks for your comment and suggestion. According to your suggestion, we have corrected the captions.

The authors stated that: “The experimental results show that the comprehensive mechanical properties of Ti(C,N)/Mo-Co-Ni/10vol%CaF2@Al2O3 cermet materials are the best.” It is impossible to conclude this without a statistical analysis of the results. The authors assume that the values are significant just because they are higher than the other materials, but does this mean something? Is this significant?

Response: Thanks for your questions. The most important index of cutting tool is hardness and the most important index of cermet material is fracture toughness. In our research, compared with the Ti(C,N)-Mo-Co-Ni cermet materials, the fracture toughness and hardness of Ti(C,N)/Mo-Co-Ni/10vol%CaF2@Al2O3 cermet materials are improved by 31.83%, 24.25%, respectively. Compared with the Ti(C,N)/Mo-Co-Ni/CaF2/Al2O3 cermet materials, the fracture toughness and hardness of Ti(C,N)/Mo-Co-Ni/10vol%CaF2@Al2O3 cermet materials are improved by 22.5%, 24.31%, respectively. Meanwhile, the flexural strength reduced by only 5.5% and 1.9%. So, we think that the comprehensive mechanical properties of Ti(C,N)/Mo-Co-Ni/10vol%CaF2@Al2O3 cermet materials are the best without a statistical analysis of the results. We hope that you will agree with us, thank you!

Lines 416-417: the authors state that: “The formation of multi-layer core-shell microstructure can effectively improve the mechanical properties of Ti(C, N)-Mo-Co-Ni cermet”. Once again this cannot be concluded without a statistical analysis of the results. Just because the obtained values are higher (slightly) does not mean the material has improved properties.

Response: Thanks for your comment and suggestion. Based on the previous question, the formation of multilayer core-shell microstructure can effectively increase the fracture toughness and hardness of cermet material, and the flexural strength is slightly reduced. Because the hardness and the fracture toughness are more important and the improvement effect is better, we think that the formation of multilayer core-shell microstructure can effectively improve the mechanical properties of Ti(C,N) cermet material.

A paragraph discussing the study limitations needs to be added to the manuscript.

Response: Thanks for your comment and suggestion. According to your suggestion, we have added the study limitations.

I suggest the paper title be changed, to more clearly describe the experimental work. For example: "synthesis and mechanical characterization of a Ti(C,N)/MoCoNi/CaF2@Al2O3 self-lubricating cermet”.

Response: Thanks for your comment and suggestion. According to your suggestion, we have changed the paper title to "Synthesis and mechanical characterization of a Ti(C,N)/Mo-Co-Ni/CaF2@Al2O3 self-lubricating cermet”.

Round 3

Reviewer 3 Report

I have carefully read the manuscript and, in my opinion, it is now suitable for publication.